# Network determinants of relationship influence on HIV prevention decision-making among people in the social networks of women who have experienced incarceration in the US

**Andrea K. Knittel**[1]*, **Gabriel Varela**[2], **Ella G. Ferguson**[1], **Hannah Hulshult**[3], **Jamie B. Jackson**[1], **James Moody**[2], **Adaora A. Adimora**[4†]

1 Department of Obstetrics and Gynecology, University of North Carolina at Chapel Hill School of Medicine, Chapel Hill, NC, United States of America, 2 Department of Sociology, Duke University, Durham, NC, United States of America, 3 University of North Carolina at Chapel Hill School of Medicine, Chapel Hill, NC, United States of America, 4 Institute for Global Health and Infectious Diseases, University of North Carolina at Chapel Hill School of Medicine, NC, United States of America

† Deceased.
* aknittel@umich.edu

**Data Availability Statement:** The datasets generated during and/or analyzed during the

## Abstract

### Background

Many cisgender women in the US who have experienced incarceration are at substantial risk for HIV acquisition after they return to the community. Various network interventions have been leveraged for HIV prevention in this population. The objective of this study was to identify network and relationship determinants of influence on HIV prevention decisions, including PrEP.

### Methods

We conducted interviews with a network mapping exercise with participants recruited from the social and sexual networks of women who had experienced incarceration. Participants enumerated important individuals in their lives from the past six months and provided demographic and relationship data as well as whether each relationship influenced their HIV prevention decisions. We abstracted network data from the interview transcripts and described the data set using descriptive statistics and network density graphs. To measure associations between characteristics at each level and whether a relationship was considered influential regarding PrEP decision-making, we use multiple logistic regression with random intercepts for each respondent.

### Results

We interviewed 32 participants, average age 33.5 years (SD = 8.98), majority female (n = 28, 87.5%), white (n = 23, 71.8%), heterosexual/straight (n = 25, 78.1%), and with a

current study are not publicly available due to the sensitive nature of the qualitative interview transcripts. The data contain sensitive information about social and sexual networks that increase the potential for deductive disclosure, and the Institutional Review Board (IRB) has determined that these may not be made publicly available. The data are available deidentified following regulatory approvals from the University of North Carolina at Chapel Hill Institutional Review Board (irb_questions@unc.edu).

**Funding:** This project was supported by joint pilot funding from the UNC Center for Health Equity Research and the Integrating Special Populations program within the UNC Clinical and Translational Science Award program of the National Center for Advancing Translational Sciences, National Institutes of Health (UL1TR002489). Dr. Knittel is a faculty scholar in the UNC WRHR Career Development Program which is supported by the Eunice Kennedy Shriver National Institute of Child Health and Human Development (K12 HD103085). Dr. Adimora was supported in part by the UNC Centers for AIDS Research funded through the National Institute of Allergy and Infectious Disease (P30 AI050410). The funders had no role in study design, data collection and analysis, decision to publish, or preparation of the manuscript.

**Competing interests:** Dr. Adimora received consulting fees from Merck and Gilead, and UNC received funds from Gilead and Merck for her research. The other authors report no conflicts of interest.

personal history of incarceration (n = 29, 90%). They reported 253 relationships (119 family, 116 friend, 18 sexual relationships). Most adult network members had used drugs or alcohol (n = 182, 80.9%), and of those, 30.8% had used them with the participant (n = 53). The mean network size was 7 (SD = 4) and network density was 52.2%. In the full model, significant positive predictors of an influential relationship included participant non-heterosexual identity (OR 27.8), older average age in the network (OR 3.9 per standard deviation), and being a current or prior sexual partner (OR 10.1). Significant negative predictors included relationships with individuals who use or had used drugs (OR 0.28), longer average relationship duration in the network (OR 0.09) and being in a network with at least one sexual partner (OR 0.2).

## Conclusions

There are significant positive and negative determinants of relationship influence related to PrEP at individual-, dyad-, relationship-, and network-levels. These support using nuanced network approaches to behavior change that respect and leverage the diversity of relationships that comprise the social networks of women who have experienced incarceration.

## Introduction

The experience of incarceration for cisgender women in the US is one of a tangled web of interconnected and interrelated biological and social factors, also called a syndemic, that produces disparities in HIV acquisition after return to the community [1–3]. The syndemic is driven by shared risk factors for HIV and incarceration (i.e., substance use and sex exchange), social and structural marginalization based on poverty, racism, and substance use, and the disruptive effects of women's incarceration on their social, sexual, and support networks [4, 5]. Despite this, initiation of pre-exposure prophylaxis (PrEP) for HIV among women who have experienced incarceration is low [6, 7]. Many of the hesitations and barriers to PrEP initiation among women with incarceration histories and women recruited for research based on other experiences are similar, including low perceived need for PrEP, stigma, cost, and medical distrust [8–10]. These similarities may result from the very high prevalence of incarceration experience among all women at heightened risk for HIV acquisition [11]. The authors recognize that both cisgender and transgender women who experience incarceration bear a disproportionate burden of HIV risk in the US. Because of the different experiences of these groups of women depending on the facilities in which they were incarcerated (i.e., "men's" facilities versus "women's" facilities, with sex at intake usually assigned based on sex assigned at birth), it is important to consider them separately. This work focuses on individuals recruited by cisgender women from their networks.

Cisgender women's social and sexual networks in the US HIV epidemic were initially conceptualized as the embodiment of the social and structural risk factors that increased women's risk for HIV transmission [4, 12]. Research on women's incarceration and their sexual networks shows that relationship disruption due to incarceration may produce concurrent partnerships, or partnerships that overlap in time, and new partnerships following incarceration [13–16]. Subsequent work has emphasized the potential of social and sexual networks to serve as necessary conduits for information and support for PrEP [17–19].

Network interventions that have been leveraged for HIV prevention for cisgender women who have experienced incarceration include: 1) Change agent, where influential individuals in the network promote behavior change; Dauria *et al* [19] have developed a peer-led patient navigator PrEP linkage intervention for women at risk for HIV acquisition leaving jails in San Francisco. 2) Segmentation, where groups are identified to change behavior together; in the E-WORTH intervention, Gilbert *et al* [20] recruited women mandated to probation, parole, or alternative-to-incarceration programs in New York City who had a history of drug use to participate in group-based sessions focused on reducing condomless sex. 3) Induction, where stimulated peer-to-peer interactions create cascades of information diffusion/behavior change; in a PrEP demonstration project, Meyer *et al* [21] used modified respondent-driven sampling to recruit women with criminal-legal involvement and their network members to participate in the demonstration PrEP clinic.

Using the modified Social Ecological Model for PrEP, the formative work and implementation science related to these interventions has identified multiple nuanced relationship- and network-level factors that may be important determinants of PrEP uptake among cisgender women who have experienced incarceration [22]. Sexual partners, for example, have been identified as untrustworthy in several studies [8, 23] but when relationships begin in environments that support more open communication, sexual partners can be seen as influential and potentially supportive [9]. Other supportive relationships that may be influential for PrEP include peers in treatment and some types of family relationships, in stark contrast to relationships in drug use networks with individuals who may be seen as high-risk and indifferent to HIV prevention strategies [9].

Given the themes related to relationships and networks that emerged in our prior work, we sought in this study to conduct a network analysis of networks reported by individuals recruited by cisgender women who had incarceration histories. We aimed to identify determinants of influence on HIV prevention decisions, including PrEP. We hypothesized that sexual partnerships and family relationships would be influential, while relationships that included substance use would not be influential.

## Materials and methods

### Source interviews

Between June 1, 2020 and December 31, 2020 we used a two-step recruitment process to identify individuals within the social networks of women who had experienced incarceration. The full recruitment and interview methodology is reported elsewhere [9]. Briefly, we partnered with three community-based organizations (CBO's) in the urban and suburban Southeastern US who provide re-entry services and/or residential substance use treatment. We first identified recruiters–women who had experienced incarceration, English-speakers, at least 18 years-old, and involved with a partner CBO. Each recruiter could recruit up to three unique network members as participants in the study–referred participants were required to be sexual partners, drug-use partners, and/or treatment partners. We did not specify any sex/gender limitations on the individuals who were referred. Most recruiters only referred 1–2 participants. Eligible participants were over the age of 18, English-speakers, not currently incarcerated, residing in North Carolina, and provided study staff the unique ID number from their recruiter. Individuals who were currently under community supervision or detained at the CBO as a condition of parole or probation were considered non-voluntary CBO participants and excluded.

Eligibility screening and informed consent were performed by one member of the research team (JJ). Verbel consent was obtained and documented in the participant log. Another member of the research team (EGF), blinded to all identifiers, then conducted all of the interviews

by phone. Demographic data were collected for the participant. The interviews started with an egocentric network (the ego-network) mapping exercise about the participant (the ego) and the important individuals in their lives (alters) over the past six months [24]. The interviewer asked about demographic data for the network alters and HIV prevention influence and PrEP support in those relationships over the prior six months. The remainder of the interview focused on prompts about individual and network HIV risk, awareness of PrEP as an HIV prevention intervention, and barriers and facilitators to PrEP uptake. For participants unaware of PrEP, at the start of the interview PrEP was briefly described as a daily oral medication that could prevent HIV, the only available formulation at the time of the interviews. The interview guide is shown in S1 Appendix. Interviews were recorded in their entirety. Each participant received a $25 gift card or a gift bag of products valued at $25 at the conclusion of the interview, depending on whether their treatment program structure precluded receiving gift cards.

The study was approved by the Institutional Review Board at the University of North Carolina at Chapel Hill (#20–0219).

## Network data

Network data were abstracted from the ego-network mapping exercise and entered into a REDCap database without identifiers. Each ego was linked in the data set to their reported network alters.

## Measures

**Influence on HIV prevention.** For each dyad, respondents were asked "How does [alter name] affect the things you do to prevent HIV?" The qualitative responses were coded to a binary variable indicating whether the relationship is influential or not. Responses such as "[alter name] does not" or "Not at all" were considered negative responses. Any indication of influence was coded as a positive response. Notably, we did not code for the valence of the influence (i.e., whether the relationship was supportive of HIV prevention activities versus discouraging of HIV prevention activities).

**Individual characteristics.** Individual-level ego characteristics were identified at the start of the interviews, including site of recruitment, age, gender, racial and ethnic identity, sexual orientation, and history of incarceration. Individual alter characteristics were abstracted from interview responses, and thus reflect the participant's perception of alter identities. Egos were asked directly about race, rural/suburban/urban residence, and substance use for each alter, and the gender of alters was assigned based on pronoun usage during the interview. Substance use was phrased as "drug use" in the interview, and alcohol was included if the ego considered the alter's use of alcohol to fall into the category of "drug use" for that alter. We did not specifically prompt the participants to consider alcohol use and, based on review of the interview transcripts, alcohol was discussed explicitly when an alter was in treatment for alcohol use disorder or if the alcohol use was considered problematic for the alter by the ego.

Interview responses were aggregated to construct relevant structural features of participants' social networks at three levels, dyad, relationship, and network.

**Dyad characteristics.** Dyad-level variables *compare* characteristics of the participants (egos) and their network alters (e.g., whether the dyad shares the same race). Dyad-level variables were constructed as binary variables indicating similarity or dissimilarity on each of the individual characteristics of age and gender. These were coded as a match or not between the ego and each of the alters' characteristics.

**Relationship characteristics.** Relationship-level variables explore the *qualities of the connection* in a given dyad (e.g., how long the dyad has known each other). Relationship-level

measures of sexual relationship history, experience with sharing drugs, duration of relationship and type of relationship (i.e., family or friend) were abstracted from participant interview responses. Sexual relationship and history of substance use together were binary variables indicating the presence or absence of each type of relationship. Types of relationships were first coded as friendships or family relationships. Family relationships were then further parsed into parental/guardian relationships if the duration of the relationship was equal to the age of the ego; as a child or younger sibling relationship if the alter was under 18 years old; or as another family member if neither of the previous categories were true. Relationship duration was abstracted in the unit of time reported by the participant and transformed to years for analysis.

**Network characteristics.** Network-level variables encompass the ego's entire ego-network such as overall composition (e.g., proportion of alters who are family members) or structure (e.g., network size, density).

Network-level measures were aggregated to reflect the composition of the ego-network as a whole. The proportion of network alters who were family members or friends was defined as number of family or friend alters over the total number of alters. The average relationship duration and average alter age in the network were calculated as means with standard deviations. We also used a binary indicator of the presence or absence of a sexual partner in the network as a whole. Network size was a count of all of the alters in a given ego-network. Network density was defined as the number of existing relationships within the ego-network out of all possible relationships within the ego-network. Relationships between alters were assessed during the qualitative interview by asking "Is this person connected to any of the other people we've listed? How?" Any affirmative response was coded as an existing relationship.

## Missing data

Transcripts were reviewed an additional time if any data elements were missing. For those where a response could not be located, the variable was left as missing.

## Analytic strategy

Descriptive tables were created to report the frequency and proportions of categorical variables and averages and standard deviations of continuous variables. To generate the network graphs, we utilized the NodeXL plugin for Microsoft Excel [25], which allows users to input data from quantitative or qualitative sources and create network visualizations.

To measure associations between characteristics at each level and whether a relationship was considered influential regarding PrEP decision-making, we use a hierarchical logistic regression approach with random intercepts for each respondent using the "lme4" package in R version 4.0.3 [26]. Continuous variables in the model are centered and scaled, so that coefficients represent a one standard deviation from the mean.

As in other network research, the structures being studied here are inherently endogenous and reciprocal [27]. Personal networks are naturally clustered at two levels: the participant ("ego") who is observed directly, and their network members with their associated connections ("alters" and "relationships"). Because sets of alters are nominated by the same ego, we can expect that these sets of alters may share observed and unobserved characteristics, and that perceptions of influence will be correlated among relationships within each ego. These correlations pose a problem for traditional modeling which assumes that observations of the dependent variable are independently distributed. However, multilevel modeling [28], and more specifically hierarchal multilevel models [29, 30] account for the nested structure of personal networks [31], where the alters can be specified as one level, nested within the higher level of

the egos. Hierarchal multilevel models introduce additional coefficients with a random component that can vary both the intercept and slope of each cluster. The random component allows the model to vary the starting intercept for each ego, reflecting a different starting probability of reporting a relationship as influential. In addition to making the effect of observed variables more interpretable, this approach has the added benefit of explaining how much of the variation in the outcome is explained by differences between egos (the differences in starting points) in contrast to the variation attributable to explanatory variables. To confirm the appropriateness of a multi-level modeling strategy, we ran a series of candidate hierarchal multilevel models compared to a baseline model where intercepts and slopes were fixed, and compared differences in fit using Akaike Information Criterion corrected for small samples (AICc) (S1 Table) [32]. We found that varying intercepts at the respondent level best fit our data.

This modeling approach has two main benefits over other popular alternative network models that account for the interdependence of observations in a network setting–quadradic assignment procedures (QAP) and network autocorrelation models. QAP are a non-parametric approach that tests significance between two or more *dyad* or *relationship* variables (e.g, shared race, closeness, time known). QAP does not allow for simultaneous tests of associations between network, dyad, relationship and individual-level variables. Network autocorrelation models are useful when there is a concern for cross-ego autocorrelation (i.e. egos within a bounded sociocentric network). In our case, the assumption that each network represents the independent, local view of an ego is more reasonable because the data were collected as ego-networks rather than from within a single larger network, and because many of the egos were recruited from different sites, the networks include multiple types of connections (e.g., family, friends), and most recruiters only recruited 1–2 participants. The hierarchical logistic model selected for this study is supported by current methodological reviews of ego-network analysis [see, for example, 33] because it enables both a contrast of variables across multiple levels, and a contrast of *within* and *between* ego variation.

The final model included the ego characteristics of age, race, gender, and recruitment site. History of incarceration was not included as there were very few individuals who had not been incarcerated and specific recruiter was not included as most recruiters only recruited 1–2 participants. It included alter characteristics of age, race, pronouns, and substance use, dyad characteristics of race concordance and age difference, relationship characteristics of prior substance use together, sexual relationship, relationship type (friend, parent/guardian, child, other family), and relationship duration. It also included network characteristics of network size, average age, presence of a sexual relationship (yes/no), proportion of family in the network, average relationship duration.

We recognized that our small data set was likely insufficient for the extensive hypothesis testing that our *a priori* theoretical underpinning would recommend. That is, including the constructs from the modified Social Ecological Model for PrEP would result in a large number of multiple comparisons. Rather than potentially overcorrecting for this issue, however, we elected to report coefficients and 95% confidence intervals for comparisons made in the model and encourage consideration that the direction of identified trends should be the focus of interpretation, while the magnitude of these trends may be specific to our particular participants.

## Results

### Descriptive analysis

**Individual ego characteristics.**    A total of 32 participants were interviewed. Individual characteristics are shown in Table 1. Their average age was 33.5 years (SD = 8.98), and the

**Table 1. Full sample and analytic sample ego, alter, dyad, and network characteristics among participants recruited from the social networks of women who have experienced incarceration in the Southeastern US, 2020.**

| | Full sample | Analysis sample |
|---|---|---|
| | N = 252 | N = 216 |
| | N(%) | N(%) |
| Site: Site 3 | 215 (85.3%) | 186 (86.1%) |
| Influential relationship: Yes | 132 (53.7%) | 117 (54.2%) |
| Ego age | 34.7 (9.39)* | 34.3 (9.31)* |
| Ego sex: Male | 27 (10.7%) | 20 (9.26%) |
| Ego sexual orientation: Hetero | 210 (83.3%) | 179 (82.9%) |
| Non-hetero | 42 (16.7%) | 37 (17.1%) |
| Ego race: Black | 58 (23.0%) | 43 (19.9%) |
| Other | 12 (4.76%) | 12 (5.56%) |
| White | 182 (72.2%) | 161 (74.5%) |
| Alter has used drugs | 181 (72.4%) | 156 (72.2%) |
| Alter race: Black | 54 (21.4%) | 45 (20.8%) |
| Other | 19 (7.54%) | 12 (5.56%) |
| White | 179 (71.0%) | 159 (73.6%) |
| Alter gender: man | 86 (35.7%) | 79 (36.6%) |
| Alter is or has been a drug partner | 52 (21.7%) | 48 (22.2%) |
| Alter is or has been a sex partner | 22 (8.87%) | 19 (8.80%) |
| Relationship: Child/Younger family relation | 28 (11.3%) | 25 (11.6%) |
| Friend | 133 (53.8%) | 111 (51.4%) |
| Other Family | 19 (7.69%) | 19 (8.80%) |
| Parent/Guardian | 67 (27.1%) | 61 (28.2%) |
| Ego-alter absolute age difference | 13.4 (12.4)* | 13.5 (12.5)* |
| Ego and alter are the same race | 194 (77.0%) | 169 (78.2%) |
| Relationship duration | 12.9 (15.1)* | 13.5 (15.4)* |
| Proportion of family alters in the network | 0.47 (0.14)* | 0.47 (0.15)* |
| Average relationship duration with alters in the network | 12.9 (4.66)* | 12.9 (4.88)* |
| Average alter age in the network | 38.2 (8.56)* | 37.9 (8.31)* |
| Network contains at least one sexual partner | 179 (71.0%) | 147 (68.1%) |
| Size of network (dyad-level) | 8.70 (2.21)* | 8.39 (2.13)* |

* mean (standard deviation)

majority reported female sex (n = 28, 87.5%) and heterosexual/straight sexual orientation (n = 25, 78.1%; homosexual/lesbian/gay/bisexual n = 4, 12.5%). Self-reported racial/ethnic identities included Black/African American ("Black," here forward; n = 6, 18.8%) and white (n = 23, 71.8%), with the additional participants reporting another racial/ethnic identity (n = 3, 9%); 1 participant reported more than one race/ethnicity. Among the participants, 29 (90%) had a personal history of incarceration, either in jail (n = 21, 72%) or prison (n = 8, 28%). Of those with a history of incarceration, 12 (41.2%) were within the past year, 5 (17.2%) were during the prior year, and 19 (65.5%) were before the prior year.

**Individual alter characteristics.** Participants reported 253 unique alters (mean number of alters per participant = 7.9, SD = 2.5), including 119 family members, 116 friends, and 18

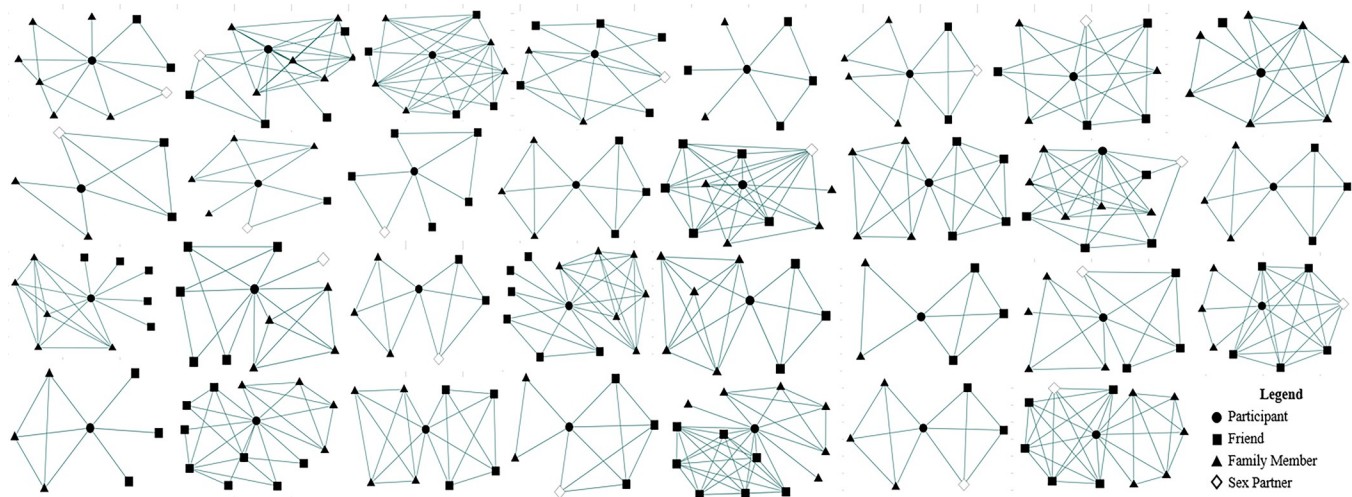

**Fig 1. Ego-network graphs for participants recruited from the social networks of women who have experienced incarceration in the Southeastern US, 2020.**

sexual partners. Participants reported 80.9% (n = 182) of adult alters had used drugs or alcohol, either current or past use. Among friend alters, 95.7% (n = 111) had used drugs either currently or previously; among family alters, 60.0% (n = 55), and among sex partner alters, 88.9% (n = 16).

**Dyad characteristics.** Comparing egos with their alters, 78.8% of friend dyads and 70.6% of sexual partners were racially concordant relationships; multiracial participants were counted as discordant.

**Relationship characteristics.** The mean duration for friend relationships was 1.9 years (SD = 3.2) years and for sexual relationships was 4.0 years (SD = 10). Among relationships with alters who used drugs either currently or in the past, 30.8% (n = 53) had used with the participant. Among alters who had used drugs, 9.9% (n = 11) of friends, 70.9% (n = 39) of family, and 18.8% (n = 3) sexual partners had used with the participant, respectively.

**Ego-network characteristics.** Network density graphs are shown in Fig 1. The mean ego-network size was 7 (SD = 4) averaged over individuals. The average ego-network size averaged over dyads is shown in Table 1. The mean numbers of friends and family members were 3 (SD = 2) and 4 (SD = 2), respectively. The average network density was 83.8% (SD = 0.2) within friend subnetworks, 83.4% (SD = 0.3) within family subnetworks, and 52.2% (SD = 0.1) overall.

**Analytic sample.** Of the reported alters and relationships, 37 had missing data and were excluded; this resulted in complete exclusion of one participant. Of those 37, 11 are missing information about the alter's gender, 12 are missing information about the alter's drug use, 10 are missing information about the alter's place of residence, 13 are missing information about the alter's age, 6 are missing information about the duration of the relationship and 6 are missing information about the influence of the relationship on HIV Prep use. Table 1 displays the differences between our original sample and the final analysis sample after accounting for missing data. There are no significant differences between the full and analysis sample, although the analysis sample contains somewhat fewer observations from Black respondents and about Black alters. As such, our final sample includes 31 respondents and 216 relationships, with an average number of relationships per respondent of 7 (SD = 1.4). Among the 216 observations, 196 (90.74%) were reported by female respondents, and the average respondent age is 34.3 years (SD = 9.31). There were 19 (8.80%) sexual relationships reported and 48

**Table 2. Bivariate relationships between ego, alter, dyad, and network characteristics and relationship influence on PrEP and HIV prevention decision-making among participants recruited from the social networks of women who have experienced incarceration in the Southeastern US, 2020.**

| | No | Yes | p-value |
|---|---|---|---|
| | N = 99 | N = 117 | |
| | N(%) | N(%) | |
| Ego age (years) | 33.5 (8.52)* | 35.0 (9.91)* | 0.258× |
| Ego sex: Male | 10 (10.1%) | 10 (8.55%) | 0.875+ |
| Ego sexual orientation: | | | 0.002+ |
| Hetero | 91 (91.9%) | 88 (75.2%) | |
| Non-hetero | 8 (8.08%) | 29 (24.8%) | |
| Ego race: | | | 0.015+ |
| Black | 13 (13.1%) | 30 (25.6%) | |
| Other | 3 (3.03%) | 9 (7.69%) | |
| White | 83 (83.8%) | 78 (66.7%) | |
| Alter has used drugs | 76 (76.8%) | 80 (68.4%) | 0.223+ |
| Alter race: | | | 0.471+ |
| Black | 17 (17.2%) | 28 (23.9%) | |
| Other | 6 (6.06%) | 6 (5.13%) | |
| White | 76 (76.8%) | 83 (70.9%) | |
| Alter gender: man | 33 (33.3%) | 46 (39.3%) | 0.443+ |
| Alter is or has been a drug partner | 22 (22.2%) | 26 (22.2%) | 1.000+ |
| Alter is or has been a sex partner | 4 (4.04%) | 15 (12.8%) | 0.042+ |
| Relationship: | | | 0.849+ |
| Child/Younger family relation | 10 (10.1%) | 15 (12.8%) | |
| Friend | 52 (52.5%) | 59 (50.4%) | |
| Other Family | 10 (10.1%) | 9 (7.69%) | |
| Parent/Guardian | 27 (27.3%) | 34 (29.1%) | |
| Ego-alter absolute age difference (years) | 13.9 (13.5)* | 13.2 (11.7)* | 0.661× |
| Ego and alter are the same race | 81 (81.8%) | 88 (75.2%) | 0.314+ |
| Relationship duration (years) | 13.1 (14.6)* | 13.8 (16.1)* | 0.728× |
| Proportion of family alters in the network | 0.47 (0.18)* | 0.47 (0.12)* | 0.708× |
| Average relationship duration with alters in the network (years) | 12.8 (5.25)* | 13.0 (4.56)* | 0.847× |
| Average alter age in the network (years) | 36.6 (6.73)* | 39.0 (9.34)* | 0.031× |
| Network contains at least one sexual partner | 75 (75.8%) | 72 (61.5%) | 0.037+ |
| Size of network (dyad-level) | 8.45 (1.87)* | 8.34 (2.33)* | 0.694× |

\* mean (standard deviation)

× result of t-test

+ result of chi-square test

(22.2%) relationships that included drug sharing behavior at some point during the relationship.

**Bivariate analysis.** In our analysis sample, 117 (54.2%) of relationships are perceived as influential by respondents.

Table 2 illustrates the difference between influential and non-influential relationships along a set of individual- and network-level characteristics. At the individual level, influential relationships were more prevalent among non-heterosexual respondents (n = 8, 8.08% of non-influential relationships versus n = 29, 24.8% of influential ones). Influential relationships were also more prevalent among individuals who identified as "Black" or "other race" in contrast to

those who identified as "white", with Black respondents reporting 13.1% (n = 13) of non-influential relationships compared to 25.6% (n = 30) of influential relationships. Sexual partner relationships were more likely to be influential (n = 4, 4.04% non-influential sexual relationships vs n = 15, 12.8% of influential sexual relationships). At the network level, the average age of alters in networks containing influential relationships was also significantly higher (mean 35 [SD = 9.91] vs 33.5 [SD = 8.52]).

**Logistic regression analysis.** In Fig 2, we report the log-odd results of the hierarchical logistic regression analysis predicting whether a respondent reports a relationship as influential when it comes to PREP attitudes. Model coefficients are available in S2 Table where we report both log-odds and odds ratio. We report odds ratios in the text. Many of the relationships highlighted by independently comparing influential and non-influential relationships remain robust to the introduction of random intercepts and controls. The full model included all individual- and network-level controls.

For individual ego characteristics, non-heterosexual individuals had 27.87 times the odds of heterosexual individuals to report a relationship as influential. The disparity between Black and white respondents is no longer significant in the full model.

There are several significant alter and relationship characteristics. Being in a relationship with an individual alter who uses or has used drugs resulted in 0.28 times the odds of that relationship being influential. Relationships that are or have been sexual partnerships had 10.09 times the odds of non-sexual relationships of being influential. A standard deviation increase in the average relationship duration in the network was also associated with 0.09 times the odds of reporting influence.

At the network level, while we found that sexual partnerships increased the probability of finding a relationship influential, our model also suggests that relationships embedded within a network that contains at least one sexual partnership have 0.21 (p < 0.1) times the odds of finding relationships influential *in general* when compared to relationships embedded in networks without a sexual partner. This significantly dampens the beneficial impact of a relationship with a sexual partner, and highlights the importance of teasing out difference network levels independently. Relationships that were embedded in networks whose average alter age is one standard deviation above the mean have 3.92 times the odds of being influential.

## Discussion

This statistical analysis of network determinants of relationship influence on PrEP decision-making among people in the social networks of women who have experienced incarceration produced several significant insights. First, the participants we recruited from the networks of cisgender women with incarceration experience were themselves primarily women who had experienced incarceration. The ego-networks were largely comprised of individual alters of the same race as the participant and most friends and family members within the networks had histories of substance use. Second, some relationships and relationships in some networks were significantly more likely to be influential in the participant's decision-making about HIV prevention and PrEP. Relationships in networks reported by non-heterosexual participants or older participants and relationships in networks with older network members were more likely to be influential. Additionally, sexual relationships had two effects. At the relationship level, sexual relationships themselves were more likely to be influential. At the network level, the presence of a sexual relationship in a network reduced the likelihood that other, non-sexual relationships in that network would exert influence. Finally, relationships with individuals who use or had used drugs were much less likely to be influential.

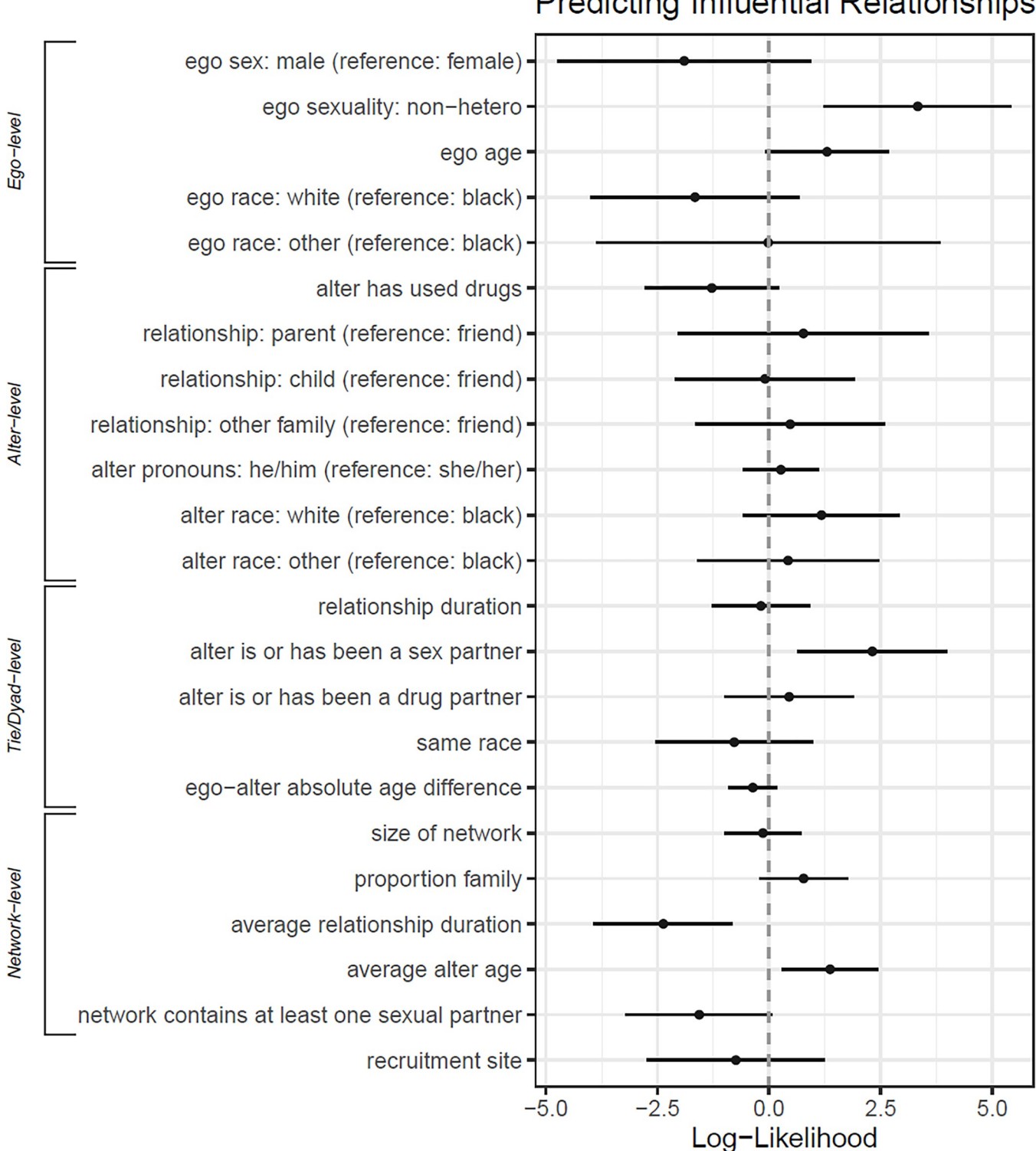

**Fig 2. Adjusted log-odds for ego, alter, dyad, relationship, and network characteristics and relationship influence on PrEP and HIV prevention decision-making among participants recruited from the social networks of women who have experienced incarceration in the Southeastern US, 2020.**

These network and relationship-level results affirm and extend prior analyses from this and other samples of women with criminal legal involvement and their networks. The importance of sexual partners in HIV prevention and decision-making around PrEP reflects the prior qualitative findings from this sample and others describing the importance of communication and trust in sexual relationships [9] and the potential need for discreet methods of HIV prevention with untrustworthy partners [8, 23]. Further, our finding that relationships with individuals who use or had used substances were substantially less likely to be influential was consistent with reports that individuals who use substances are considered to be high-risk network members who are indifferent to the HIV prevention decisions of the respondents [9, 34]. This finding also suggests that the previously described themes of supportive treatment peers and supportive substance use disorder treatment environments may have been driven more by the structure and community-building in the treatment program rather than a more general shared experience of substance use. Finally, the effect of age–both of older respondents and of networks of older people–is a novel one in this population. It contrasts somewhat with an earlier qualitative finding that participants experienced their children as important motivators [8]. This suggests that parenting relationships with young people (i.e., children and other young relatives) are highly salient for a small subgroup, such as those mothering infants or school-age children, whereas respected older family members and peers are more likely to be influential overall.

This analytic approach is not without limitations. Although social network recruitment, or snowball sampling, is an accepted methodology for reaching participants from marginalized groups, recruiters for this study may have selected participants to refer based on their real or perceived interest in or openness to the topic of HIV prevention [35, 36]. Additionally, this approach resulted in a potentially heterogeneous sample of network members. Most substantively, this network analysis was undertaken with ego-network data collected during qualitative interviews. As a result, the sample size is substantially less than that of traditional quantitative network analysis. This resulted in large magnitude coefficients that may imply some skewness in our data and large standard errors that urge careful interpretation of the findings. Since we also conducted a series of model fit robustness checks that compare models of increasing complexity and find substantial support for our model according to well-established cutoffs [32], we believe that the benefits of this mixed-methods approach outweigh the limitations. We suggest that the trends that are revealed by our study have important implications for interventions with women who have experienced incarceration and their networks–while the magnitude of these trends are specific to our particular data sources.

Additionally, since in ego-network collection, the ego, alter, and relationship characteristics are all reported by the ego, some caution is warranted in interpreting the model results as network alters themselves might report different identities and experiences. We also assessed alter substance use via the report of the ego, so it is possible that some substance use was unknown to the ego and that alcohol use, in particular, may not have been included as substance use if the ego did not perceive it to rise to the level of problematic use for the alter. Finally, we coded for the presence or absence of influence around HIV prevention in relationships, not whether that influence was positive or negative toward health promoting activities. Our qualitative analyses of these interviews do suggest that many egos understood the question about influence to be asking in particular about positive influence in support of HIV prevention; these egos responded with examples of partners, friends, and family supporting them to engage in behaviors or make decisions that would help to avoid HIV acquisition.

The results of this study also raise important questions for future research. Beyond ego-networks, efforts toward analyses of longitudinal network data, complete networks across communities or organizations, and information networks that include online alters would all provide important additional perspectives on the ways that women who have experienced

incarceration and individuals in their networks make decisions about HIV prevention. Larger network studies building on these findings could measure HIV-related knowledge, risk perception, and prevention behaviors across networks and over time to empirically understand diffusion of information and behaviors across networks that include cisgender women with histories of incarceration. Additionally, we did not address the potential for transmission of *dis-* or *mis*-information about HIV prevention within networks, although this is likely to be an important determinant of HIV prevention behaviors.

Taken together, these findings nonetheless have important implications for the implementation of network interventions focused on HIV prevention among women who have experienced incarceration and their social and sexual networks. For change agent and induction interventions, our results suggest that effectiveness of individuals selected to promote HIV-related behavior change or the particular peer-to-peer interactions that are stimulated may vary substantially based on the specifics of those relationships and shared experiences (e.g., shared experiences of incarceration versus shared history of substance use). For segmentation interventions, targeting groups of individuals to change behavior together, such as adopting PrEP or increasing condom use, without including sexual partners may be more difficult based on our finding that sexual relationships may draw influence away from the rest of the network. The potential for leveraging sexual partnerships directly is complicated by prior findings that partners may be untrustworthy and in the most extreme circumstances may be physically or emotionally abusive [8, 9, 23], but where possible, these highly influential relationships may be important conduits for behavior change.

There are also lessons to take from this study for policy approaches to eliminate HIV acquisition among women who have experienced incarceration. Policies aimed at addressing structural-level barriers to HIV prevention, such as improving insurance coverage and building clinical capacity for PrEP [37] will also need to contend with network dynamics. One way this could be operationalized would be to prioritize insurance coverage strategies that are not determined or affected by marital status either via spousal employment or income. Another would be to emphasize capacity for PrEP provision in the places that people already go to receive other services from trusted providers, such as primary care offices and sexual and reproductive health clinics [38], in order to provide another influential relationship that might counterbalance negative HIV prevention influences in individual networks.

Overall, these findings support using nuanced network approaches to behavior change for HIV prevention that respect and leverage the diversity of relationships that comprise the social networks of women who have experienced incarceration.

## Supporting information

**S1 Appendix. Interview guide.**
(PDF)

**S1 Table. Comparison of AIC values for statistical models of relationship influence on PrEP and HIV prevention decision-making among participants recruited from the social networks of women who have experienced incarceration in the Southeastern US, 2020.**
(PDF)

**S2 Table. Full model coefficients for the multi-level model of ego, alter, dyad, and network characteristics and relationship influence on PrEP and HIV prevention decision-making among participants recruited from the social networks of women who have experienced incarceration in the Southeastern US, 2020.**
(PDF)

## Acknowledgments

The authors would like to thank all the participants and CBO staff, without whom this study would have been impossible. In addition, they would like to acknowledge Drs. Hendree Jones and Kim Andringa for their formative feedback on the study design. Dr. Ada Adimora passed away before the submission of the final version of this manuscript. The authors mourn her passing and Dr. Andrea Knittel accepts responsibility for the integrity and validity of the data collected and analyzed.

## Author Contributions

**Conceptualization:** Andrea K. Knittel, James Moody, Adaora A. Adimora.

**Data curation:** Jamie B. Jackson.

**Formal analysis:** Andrea K. Knittel, Gabriel Varela, Hannah Hulshult, James Moody.

**Funding acquisition:** Andrea K. Knittel.

**Investigation:** Andrea K. Knittel, Ella G. Ferguson.

**Methodology:** Gabriel Varela.

**Project administration:** Jamie B. Jackson.

**Supervision:** Andrea K. Knittel, Adaora A. Adimora.

**Visualization:** Gabriel Varela, Hannah Hulshult.

**Writing – original draft:** Andrea K. Knittel, Gabriel Varela, Hannah Hulshult, Jamie B. Jackson.

**Writing – review & editing:** Andrea K. Knittel, Gabriel Varela, Ella G. Ferguson, Hannah Hulshult, Jamie B. Jackson, James Moody, Adaora A. Adimora.

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
