## [Decision Letter · Decision Letter 0]

18 Jun 2024

PONE-D-24-05591Network determinants of relationship influence on HIV prevention decision-making among people in the social networks of women who have experienced incarceration in the USPLOS ONE

Dear Dr. Knittel,

Thank you for submitting your manuscript to PLOS ONE. After careful consideration, we feel that it has merit but does not fully meet PLOS ONE’s publication criteria as it currently stands. Therefore, we invite you to submit a revised version of the manuscript that addresses the points raised during the review process. 

We look forward to receiving your revised manuscript.

Kind regards,

Keith Leverett Warren, Ph.D.

Academic Editor

PLOS ONE

 [This project was supported by joint pilot funding from the UNC Center for Health Equity Research and the Integrating Special Populations program within the UNC Clinical and Translational Science Award program of the National Center for Advancing Translational Sciences, National Institutes of Health (UL1TR002489). Dr. Knittel is a faculty scholar in the UNC WRHR Career Development Program which is supported by the Eunice Kennedy Shriver National Institute of Child Health and Human Development (K12 HD103085). Dr. Adimora was supported in part by the UNC Centers for AIDS Research funded through the National Institute of Allergy and Infectious Disease (P30 AI050410).].  

Additional Editor Comments (if provided):

Reviewers' comments:

Reviewer's Responses to Questions

**Comments to the Author**

1. Is the manuscript technically sound, and do the data support the conclusions?

Reviewer #1: Partly

Reviewer #2: Yes

2. Has the statistical analysis been performed appropriately and rigorously? 

Reviewer #1: I Don't Know

Reviewer #2: Yes

3. Have the authors made all data underlying the findings in their manuscript fully available?

Reviewer #1: Yes

Reviewer #2: Yes

4. Is the manuscript presented in an intelligible fashion and written in standard English?

Reviewer #1: Yes

Reviewer #2: Yes

5. Review Comments to the Author

Reviewer #1: Thank you so much for the opportunity to review this manuscript which reports on a study investigating factors that influence decisions about HIV prevention. This is a thoughtfully designed study that builds on formative work in other contexts and from prior qualitative studies examining prevention decision making. This manuscript is also beautifully written. My major question is related to the analytic approach used and its appropriateness for network data. This concern and other observations are detailed below and shared with the intention of strengthening the presentation and contribution of this work:

I appreciated the content in the methods section that explained the interdependence in the data and the need for alternative approaches. However, I am left wondering whether the most appropriate analysis approach was used here. I’m curious why logistic regression was used instead of other analytic approaches that account for this nesting in networks (e.g. network autocorrelation models, mrQAP). The authors might need to revisit the analysis. If the current analysis is indeed the most appropriate for the data (compared to other autocorrelation approaches), then it would help to expand the paragraph in the methods section explaining why.

Regarding the analysis, was the amount of time since release from incarceration assessed? The composition of ego networks, strength of ties, and influence is likely to change over time after release and might be something to control for in analysis. Similarly, did the analysis control for recruiter since participants were recruited from their own social network?

Include the interview guide so others can understand exactly what was asked and where investigators abstracted information.

Were participants asked about relationships among alters in their social network? I’m wondering how this information was gathered to allow for ego network density calculations.

The language of influence can get a little tricky and I appreciate how this study focused on ego’s reports of alters influence (yes/no) rather than the valence of the influence (positive or negative). That said, the analysis generates results that suggest that a factor is positively associated (meaning exerts more influence, not necessarily positive or negative) or negatively associated (meaning exerts less influence) - if I’m reading this right. The authors were very careful with how they wrote about this and it might be helpful to call out the emphasis on influence (rather than valence) in the discussion section.

As authors refine this manuscript it might help to keep the network-naive reader in mind by keeping language consistent (e.g. tie or dyad; ego networks) throughout the paper. Likewise, the specific dyad-level and network measures generated (and how) might be presented in a complementary table.

Out of curiosity, how does excel generate network graphs?!

There are a very small number (n=3) of participants who reported Asian or Latinx race/ethnicity and there might be privacy concerns (risk of deduced identity). The authors might consider rephrasing how they describe these participants.

Reviewer #2: This article describes an interesting mixed-methods study examining the influence of social networks at individual-, dyad-, relationship-, and network-levels on decision-making around PrEP use for 32 women who have experienced incarceration. The findings have important clinical, policy, and research implications. Overall, this study is well executed, used appropriate methodology, and is well-written. Thus, I am recommending acceptance with minor revisions with the following points requiring clarification.

General notes:

• There were some minor typos and grammar issues that did not significantly detract from the meaning of the article, but could use another round of proofreading.

• The formatting of tables and figures need to be aligned with APA style (if this is related to how the manuscript is uploaded to the site, please ignore)

• The study is inconsistently described throughout as qualitative, quantitative, and mixed-methods. I suggest that the authors describe it as mixed-methods throughout to reduce confusion.

Introduction:

• Please add details from the cited references on the underlying causes of HIV acquisition for incarcerated women to prevent simply stigmatizing the population as “incarcerated women = higher risk of getting HIV” as it reads now.

• Similarly, why is PrEP use low in this population? Why is it different from the general women's population?

• Is there research comparing the social and sexual networks of incarcerated and non-incarcerated cisgender women around decision-making and/or HIV prevention? If so, what are the differences, similarities? If not, it would be good to note this.

• What hypotheses did you have to guide your analyses? These are indicated in the third paragraph, but not clearly stated.

Methods:

• Overall, the descriptions of variable operations, the selection of methodology, and the analytic plan are well-written. They are very clear and easy to follow, even for readers who are not from the same field. The blinded interviews were a good choice to prevent bias from knowing participants’ background information.

• Although the sample size was small, it may still be worthwhile to display the number of participants who were awarded or not awarded PrEP as a characteristic if recorded. There might be differences between these two groups when asked to identify the influence of their networks on HIV prevention.

• Since you are measuring density, I’m assuming that you assessed whether dyads of alters had ties to each other as well as the ego-alter dyad. Please clarify this in the description of the network characteristics measurement.

• Were characteristics such as socioeconomic status and health insurance coverage collected in the dataset ,as these may be barrier to access PrEP?

• If using pronouns to identify 'sex', it would be better to use 'gender' instead of 'sex' as the name of the category

• “Substance use was phrased as “drug use” in the interview, and alcohol was included if the ego considered the alter’s use of alcohol to fall into the category of “drug use” for that alter.” � Did the interviewers ensure that the egos considered alcohol use as drug use? If not, this could lead to some bias in the data analysis.

• The use of terms like “higher level egos” and “lower level alters” creates confusion without clearer explanation. I am interpreting this as relating to the nesting / hierarchical structure for analysis. If this is the case, I’d suggest rewording this to be more explicit that you are describing the nesting structure.

• You specify that you are focusing on cisgender women as the egos in this study, but 12.5% of the sample reported male sex. Why were they included in the sample? What are the potential implications or complications of their inclusion?

Results:

• The results section is clearly written and the tables and figures are strong additions and aid in the interpretation of the results.

Discussion:

• “Taken together, these findings have important implications for the implementation of network interventions among women who have experienced incarceration and their social and sexual networks.” � Please specify here that the network interventions would be focused on HIV prevention behaviors

• You identify a number of important practice / intervention implications. In addition to these, what are the possible public health policy implications? What future research do you suggest that could build on your findings?

Tables / figures / supplemental material:

• Table S2: Suggest adding heading labels to indicate the coefficient, CI, SE, and OR for the table. This will help readers grasp the results more easily.

6. PLOS authors have the option to publish the peer review history of their article (what does this mean?). If published, this will include your full peer review and any attached files.

Reviewer #1: **Yes: **Alicia Bunger

Reviewer #2: **Yes: **Mer Francis, Ph.D., MSW

---

## [Author Response · Author response to Decision Letter 0]

14 Aug 2024

Please see formatted responses attached (at the end of the compiled PDF).

---

## [Editor Report · Decision Letter 1]

10 Oct 2024

Network determinants of relationship influence on HIV prevention decision-making among people in the social networks of women who have experienced incarceration in the US

PONE-D-24-05591R1

Dear Dr. Knittel,

We’re pleased to inform you that your manuscript has been judged scientifically suitable for publication and will be formally accepted for publication once it meets all outstanding technical requirements.

Kind regards,

Francisca Ortiz Ruiz, Ph.D.

Academic Editor

PLOS ONE

Additional Editor Comments (optional):

Dear Authors,

I revised the article and the comments made by the reviewers, and I can identify that the article was revised accordingly.

The reviewers' comments and the author's answers improved the article's content. Therefore, I recommend its acceptance.

Thank you very much for the opportunity to read your paper. I'm looking forward to seeing it published in the journal.

Kind regards,

Academic Editor
---

## [Editor Report · Acceptance letter]

18 Oct 2024

PONE-D-24-05591R1 

PLOS ONE

Dear Dr. Knittel, 

I'm pleased to inform you that your manuscript has been deemed suitable for publication in PLOS ONE. Congratulations! Your manuscript is now being handed over to our production team.

Kind regards, 

on behalf of

Dr. Francisca Ortiz Ruiz 

Academic Editor

PLOS ONE